Intraspecific variation in the cochleae of harbour porpoises (Phocoena phocoena) and its implications for comparative studies across odontocetes

Martins Maria Clara Iruzun maria.martins.15@ucl.ac.uk 1 2
Park Travis 2 3
Racicot Rachel 4 5 6 7
Cooper Natalie 2
1 Division of Biosciences, University College London, University of London , London , United Kingdom
2 Department of Life Sciences, Natural History Museum , London , United Kingdom
3 Department of Earth Sciences, University of Oxford , Oxford , United Kingdom
4 Forschungsinstitut und Naturkundemuseum, Senckenberg der SNG , Frankfurt am Main , Germany
5 The Dinosaur Institute, Natural History Museum of Los Angeles County , Los Angeles , CA , United States of America
6 Department of Earth and Environmental Sciences, Vanderbilt University , Nashville , TN , United States of America
7 Department of Zoology, Smithsonian Museum of Natural History , Washington , DC , United States of America
Chan Karen
Electronic publication date: 2020 Apr 13
Publication date: 2020
Volume: 8
Electronic Location ID: e8916
Received 2019 Sep 17; Accepted 2020 Mar 15
Copyright: ©2020 Martins et al.
Copyright year: 2020
Copyright holder: Martins et al.
License: This is an open access article distributed under the terms of the Creative Commons Attribution License, which permits unrestricted use, distribution, reproduction and adaptation in any medium and for any purpose provided that it is properly attributed. For attribution, the original author(s), title, publication source (PeerJ) and either DOI or URL of the article must be cited.
License URL: https://creativecommons.org/licenses/by/4.0/

Keywords: Cochlea, Intraspecific variation, Echolocation, Hearing, Harbour porpoise, Phocoena phocoena, Odontoceti

Funding: Marie Sklodowska-Curie Individual Fellowship 748167/ECHO ERC Starting Grant 677774/TEMPO This work was supported by Marie Sklodowska-Curie Individual Fellowship (748167/ECHO) to Travis Park, and ERC Starting Grant (677774/TEMPO). The funders had no role in study design, data collection and analysis, decision to publish, or preparation of the manuscript.

==============================
In morphological traits, variation within species is generally considered to be lower than variation among species, although this assumption is rarely tested. This is particularly important in fields like palaeontology, where it is common to use a single individual as representative of a species due to the rarity of fossils. Here, we investigated intraspecific variation in the cochleae of harbour porpoises (Phocoena phocoena). Interspecific variation of cochlear morphology is well characterised among odontocetes (toothed whales) because of the importance of the structure in echolocation, but generally these studies use only a single cochlea to represent each species. In this study we compare variation within the cochleae of 18 specimens of P. phocoena with variations in cochlear morphology across 51 other odontocete species. Using both 3D landmark and linear measurement data, we performed Generalised Procrustes and principal component analyses to quantify shape variation. We then quantified intraspecific variation in our sample of P. phocoena by estimating disparity and the coefficient of variation for our 3D and linear data respectively. Finally, to determine whether intraspecific variation may confound the results of studies of interspecific variation, we used multivariate and univariate analyses of variance to test whether variation within the specimens of P. phocoena was significantly lower than that across odontocetes. We found low levels of intraspecific variation in the cochleae of P. phocoena, and that cochlear shape within P. phocoena was significantly less variable than across odontocetes. Although future studies should attempt to use multiple cochleae for every species, our results suggest that using just one cochlea for each species should not strongly influence the conclusions of comparative studies if our results are consistent across Cetacea.

Introduction

Intraspecific variation, the diversity of genotypes and phenotypes within a single species, is a key component of adaptation and evolution by natural selection (O’Dell & Rajakaruna, 2011; Des Roches et al., 2018). Intraspecific variation includes variation related to size, allometry, sex, and differences related to environment or genetics. It can influence community structure and ecosystem function as much as interspecific variation (Des Roches et al., 2018). Within species, there can be variation in predator defences, parasite resistance, resource manipulation and many more, all of which can alter the number and strength of interactions, shifting the dynamics of an ecosystem (Bolnick et al., 2011). Intraspecific variation also allows local adaptation and the formation of ecotypes leading to phenotypic or genotypic divergence (Ishikawa, Onoda & Hikosaka, 2007; O’Dell & Rajakaruna, 2011).

For most morphological traits, it is assumed that degrees of intraspecific variation are generally smaller than those of interspecific variation. Nevertheless, in many studies this assumption is not (or cannot be) tested because of logistical constraints, for example, where using multiple specimens for each species is not feasible due to time or cost restrictions, or because samples are limited. Palaeontological studies, for example, commonly use a single individual as representative of a whole population or species because of the rarity of fossils (Ekdale & Racicot, 2015; Ekdale, 2016; Park et al., 2017a; Gonzales, Malinzak & Kay, 2019; Marx et al., 2018; Racicot et al., 2019; Galatius et al., 2019).

One field where this assumption has been made is the study of cetacean cochleae. The cochlea, the organ of hearing, is a complex structure with a vital role in species survival (Cantos et al., 2000). Odontocetes (toothed cetaceans) are reliant on their hearing abilities using echolocation for feeding, communicating, and navigating their habitat, among other functions (Ketten, 1992a; Ketten, 1992b). Most cross-species studies of cetacean cochlear morphology use only one cochlea for each species (e.g., Luo & Marsh, 1996; Ekdale & Racicot, 2015; Park, Fitzgerald & Evans, 2016; Churchill et al., 2016; Mourlam & Orliac, 2017) due to the rarity of specimens in natural history collections and because µCT scanning them takes time and money. Many studies also focus on using the cochlea to study interspecific variation among mammal groups (Ekdale & Rowe, 2011; Ekdale, 2013; Benoit et al., 2015; Billet, Hautier & Lebrun, 2015; Ekdale, 2016; Costeur et al., 2018; Racicot et al., 2016; Racicot, Darroch & Kohno, 2018; Galatius et al., 2019) rather than just within cetaceans. However, increasing numbers of studies are using cochlear morphology (e.g., number of turns, shape of endolymphatic sac) for phylogenetic purposes using geometric morphometrics (e.g., Billet, Hautier & Lebrun, 2015; Mennecart & Costeur, 2016; Costeur et al., 2018), which could have implications in comparative studies (Ekdale, 2010; Ekdale & Racicot, 2015; Thean, Kardjilov & Asher, 2017).

Here, we tested whether intraspecific variation is a major confounding factor in cross-species analyses by comparing morphological variation among the cochleae of 18 specimens of harbour porpoise (Phocoena phocoena) with those across 51 other species of odontocete. Specifically we investigate whether variation within P. phocoena cochleae is significantly lower than variation across all odontocete cochleae. Intraspecific variation has not been quantified relative to interspecific variation in odontocete cochleae. Previous studies have either briefly noted clustering of taxa in morphospace, differences between left and right ears (Racicot, Darroch & Kohno, 2018), or only focused on population-level differences within a single species (Schnitzler et al., 2017). Our 18 conspecific specimens represent a large advance in our ability to quantify intraspecific variation in this feature.

Material and Methods

Data collection

We chose the harbour porpoise (Phocoena phocoena) for this study because of the large number of specimens in natural history collections making it possible to obtain a large sample size for the species, something that is not available for most odontocetes. We assumed that results from P. phocoena can be generalised across odontocetes, but were unable to test this assumption with the available data.

3D shape data

TP collected the majority of the 3D shape data for an earlier paper (Park et al., 2019). 18 cochleae of P. phocoena were added to this study by MCIM (Table S2). For all cochleae we obtained µCT scans of the periotics - the bone containing the inner ear - of 52 species (comprising 94% of extant genera) of odontocetes by imaging osteological specimens from museum collections. Scan parameters for cochleae of P. phocoena are available in Table S1 and scan parameters for previously collected data on the other species are available from Park et al. (2019). For this study we obtained periotics from 18 specimens of P. phocoena. Of these 15 are from the left-hand side of the skull, and three from the right-hand side; 17 are from the UK coast, and one is from the USA coast; four are female, five are male, and nine are unsexed (see Table S3). All specimens are adult animals.

Using these data, we reconstructed 3D models of the inner ears using the segmentation and thresholding editors in Avizo 9.0 (Visualization Sciences Group-a FEI Company, 2016), and then cleaned the resulting 3D models using Geomagic Wrap® (3D Systems, 2017). Next we landmarked the digital models with 40 sliding semilandmark curves comprising a total of 361 landmarks (Fig. 1), using IDAV Landmark (Wiley, 2005). The position of these curves followed the protocols of Costeur et al. (2018), using only the curves from the cochlea because the semi-circular canals are not phylogenetically or ecologically informative in odontocetes (Costeur et al., 2018). Landmarks were placed by a single investigator (TP) to avoid multi-user bias in placement. Finally we exported coordinates from the landmarked models as .pts files from IDAV Landmark (Wiley, 2005).

Figure 1 Endocast of NHMUK_SW1934.31 in anterior, lateral, dorsal and ventral views showing semilandmark curves.

(A) Anterior view, (B) lateral view, (C) dorsal view, (D) ventral view. Dotted lines show the semilandmark curves.

Geometric morphometrics

We performed all geometric morphometric analyses in R version 3.4.3 (R Core Team, 2019), using the R package GEOMORPH (Adams, Collyer & Kaliontzopoulou, 2018). First, we used Generalised Procrustes Analysis (GPA) to remove the effects of position, scale and orientation. The semilandmarks were ‘slid’ along their tangent vectors between adjacent semilandmarks until their positions minimised the shape difference between specimens (using the Procrustes distance criterion), to reduce the effect of their initially arbitrary placement (Bookstein et al., 1999; Gunz, Mitteroecker & Bookstein, 2005; Adams, Rohlf & Slice, 2013). We then performed a principal component analysis (PCA) on the resulting Procrustes coordinates using the ‘plotTangentSpace’ function; and used these principal components (PCs) in further analyses.

Linear measurements

One of us (MCIM) took the following measurements of the cochlea (Fig. 2) in Avizo 9.2.0 using the Measure and Spline Probe tools and following protocols from Park, Fitzgerald & Evans (2016): (1) cochlear height; (2) cochlear width; (3) number of turns of cochlear canal; (4) cochlear volume; (5) cochlear canal length; (6) length of secondary spiral lamina (SSL); (7) basal ratio (cochlear height/cochlear width); (8) axial pitch (cochlear height/number of turns); and (9) percentage of extent of secondary spiral lamina ((secondary spiral lamina length/cochlear canal length)*100). We also measured (10) width of basal turn perpendicular to cochlear width (W2); (11) maximum distance between turns of the cochlea (ITD); and (12) area of fenestra cochlearis (FC) following Churchill et al. (2016). Where possible (n = 7) for the specimens of P. phocoena, we also collected condylobasal length (skull length in mm) as a proxy for body size as body mass data were not available for all specimens.

Figure 2 Line drawing of Phocoena phocoena (NHMUK_SW1934.31) cochlea in vestibular and posterior view, illustrating key measurements.

(A) vestibular view, (B) posterior view.

We chose these measurements because they relate to auditory function in cetaceans. Cochlear height (1) and width (2, 10) relate to the auditory abilities of odontocetes (West, 1985; Ketten & Wartzok, 1990). The number of turns of the cochlear canal (3) also relates to auditory abilities and differs among mammal groups (West, 1985; Ketten & Wartzok, 1990; Ekdale & Rowe, 2011). Cochlear volume (4) is used as an estimate of size which relates to body mass and length (Racicot et al., 2016). Cochlear canal length (5) correlates with body size in odontocetes and is used as a quantitative measurement for comparison (Ketten, 1992a). Length of secondary spiral lamina (6) was measured to calculate percentage of extent of secondary spiral lamina (9), an indicator of the stiffness of the basilar membrane that is associated with high frequency hearing (Ekdale & Racicot, 2015; Park et al., 2017a). Basal ratio (7) and axial pitch (8) are two quantitative ways to examine cochlear shape and size for comparative studies (Racicot et al., 2016). Both measurements are negatively correlated with frequency of hearing (Ketten & Wartzok, 1990). Taxa that use high frequency hearing also tend to have a larger inter-turn distance (11; Ekdale & Racicot, 2015). The area of the fenestra cochleae (12) tends to be larger in low frequency species, although beaked whales and sperm whales also have large areas.

We then performed a principal component analysis (PCA) on the scaled and centred linear measurements listed above using the ‘prcomp’ function in R; and used these principal components (PCs) in further analyses.

All data used in this study are available on the NHM Data Portal (https://data.nhm.ac.uk/). The 3D shape data for cochleae of P. phocoena, linear measurements of all cochleae, and other data used in this paper are deposited in Martins, Park & Cooper (2019). 3D shape data for cochleae of other odontocetes were collected for Park et al. (2019) and are deposited in Park et al. (2018).

Analyses

All analyses were carried out in R version 3.4.3 (R Core Team, 2019), and reproducible R scripts are available on GitHub (https://github.com/nhcooper123/intraspecific-porpoise; Cooper, Martins & Park, 2020).

Quantifying intraspecific variation in Phocoena phocoena

To quantify intraspecific variation in cochleae of P. phocoena we estimated disparity for all principal components of the 3D shape data and linear measurements separately, using the R package dispRity (Guillerme, 2018). We used the sum of variances and median centroid distance metrics as our disparity metrics. For each linear measurement we also calculated the coefficient of variation (CV), i.e., (standard deviation/mean)*100. Generally values of CV below 30% represent low levels of variation (Brown, 1998), and previous studies that found low levels of intraspecific variation in cochleae find values of CV below 20% (Ekdale & Rowe, 2011; Mennecart & Costeur, 2016; Racicot, Darroch & Kohno, 2018). We also calculated values of CV for our measurements using data from previous studies of cochleae intraspecific variation in odontocetes so we could compare the values to those of P. phocoena. Where these studies contained right and left cochleae from one individual we first took the mean for the individual, then calculated CV across individuals.

Comparing intraspecific variation and interspecific variation

Quantifying the amount of intraspecific variation alone does not tell us whether intraspecific variation may confound the results of studies of interspecific variation. To investigate this we tested whether there were significantly higher levels of variation in cochlear shape among P. phocoena and the other odontocetes than within the specimens of P. phocoena. For the 3D shape data, we first conducted a Procrustes MANOVA (multivariate analysis of variance) using the ‘proc.lm’ function from GEOMORPH using the Procrustes aligned coordinates as the response variable, and the grouping variable (P. phocoena vs. other odontocetes) as the explanatory variable. Next we fitted a standard MANOVA using the ‘manova’ function in R, with the first 26 PC axes scores (comprising 95% of the variance in cochlear shape) as the response variable. To identify which PCs, if any, were most important in driving any differences among the cochleae of P. phocoena and other odontocetes we also used analysis of variance (ANOVA) to test each PC separately. To account for multiple testing we Bonferroni corrected our p values. Finally, we repeated the latter two analyses, i.e., MANOVA and ANOVAs, using the first six PC axes (comprising 95% of the variance) extracted from the linear measurements.

We visualised specimen position in 2D morphospace across combinations of PCs 1, 2, and 3 (3D shape data) and PCs 1–4 (linear measurement data) as these PCs were significant in our ANOVAs (see Results). To aid the interpretation of our results we also created morphospace plots of PC1 and PC2 where species were coloured based on taxonomic family and a selection of ecological variables; specifically habitat (riverine ≥ 50% of time spent in river, nearshore ≥ 50% of time spent in coastal waters, oceanic ≥ 50% of time spent in pelagic waters), diet (generalist = consumes a wide variety of prey types, fish ≥ 50% of diet is fish, cephalopods ≥ 50% of diet is cephalopods), feeding mode (raptorial = prey capture occurs mainly via snapping or ram feeding sensu Hocking et al., 2017, suction = prey capture occurs mainly via suction, i.e., obligate suction feeders), and dive type (shallow = maximum estimated dive depth less than 100 m; middle depth = maximum estimated dive depth ∼500 m; deep = maximum estimated dive depth ∼1,000 m; very deep = maximum estimated dive depth greater than 1000 m) and hearing type (NBHF = known or presumed to use narrow band high-frequency sonar; Type 2 = do not use NBHF). Some taxa fall into several of these categories so we assigned these to composite categories e.g., Tursiops is classed as nearshore/oceanic. Details of the categories for each species, and the references these were taken from, are in Tables S4 and S5. We also tested whether these variables were correlated with overall cochlear shape using MANOVAs.

Sensitivity analyses

Because we have 18 specimens of P. phocoena and only one specimen of each of the other species, cochleae of P. phocoena will have a stronger effect on the principal components scores than other cochleae, distorting the morphospace. Ideally we would collate an equal number of cochleae for each species, but these specimens simply do not exist in these numbers for the majority of species. Specimens of P. phocoena are the most readily available because they are common and often strand (Coombs et al., 2019). Their skulls are also small enough to store in bulk within natural history collections. To determine whether this distortion problem influenced our conclusions, we sampled two specimens of P. phocoena from the 18 and then repeated the GPA, PCA, and MANOVA analyses described above for the 3D shape dataset. We repeated this for every possible combination of two specimens of P. phocoena, and then for every combination of three specimens and so on until reaching every combination of 17 specimens.

To determine whether our results apply only to broad-scale taxonomic comparisons, i.e., P. phocoena vs. other odontocetes, we also repeated our GPA, PCA and MANOVA analyses comparing P. phocoena with other species of Phocoena only (P.dioptrica, P.sinus, P.spinipinnis). 3D shape analyses used PC1 - PC14, and linear measurement analyses used PC1 - PC7 as these were the PCs required to account for 95% of the variation in cochlear shape in each case.

Finally, variation within cochleae of P. phocoena may be explained by the differences among the specimens (Table S3) in terms of whether the cochleae came from the left or right side of the animal, their sex, origin or body size. We therefore tested whether variation in cochleae of P. phocoena was significantly correlated with the side of the head the cochlea came from, or the sex, origin (UK vs. USA) or skull length (as a proxy for body size) of the specimens. We repeated our GPA and PCA analyses for the specimens of P. phocoena only, and then used the resulting Procrustes coordinates or PCs as response variables in Procrustes ANOVAs or MANOVAs with either side, sex, origin, or log skull length as predictors. 3D shape analyses used PC1 - PC16, and linear measurement analyses used PC1 - PC7 as these were the PCs required to account for 95% of the variation in cochlear shape in each case. For 3D shape analyses where sex or skull length were the predictors we used only PC1 - PC6 (accounting for 75% of the variation on cochlear shape) because there were not enough observations to fit MANOVAs for all 16 PCs.

Results

3D shape variation

The first 26 principal components (PC) axes comprised 95% of the variance in cochlear shape. PC1 accounted for 34.08% of the cochlear shape variation (Fig. 3), and represents cochleae that are positively correlated with having a less oval-shaped and more circular fenestra vestibuli (Fig. 4). PC2 accounted for 12.15% of the shape variation and represents cochleae that are negatively correlated with having a radial expansion of the scala tympani (i.e., a tympanal recess, sensu Park et al., 2017b; Park, Fitzgerald & Evans, 2017; Figs. 3 and 4), and PC3 accounted for 8.956% of the shape variation (Fig. 3), and represents cochleae that are positively correlated with having cochlear canals that do not overlap the basal turn.

Linear measurements

The first six PC axes comprised 95% of the variance in cochlear linear measurements. PC1 accounted for 54.28% of the variation (Fig. 5), which is best described by a combination of cochlea width, W2, ITD and cochlea height. PC2 accounted for 18.65% of the variation best described by basal ratio and extent of secondary spiral lamina (Fig. 5). PC3 (8.579% of the variation) and PC4 (6.355% of the variation) are best described by secondary spiral lamina extent and number of turns respectively (Fig. 5).

Figure 3 Principal components (PC) plots for 3D cochlea shape data for PCs 1–3.

Purple points and convex hulls are Phocoena phocoena specimens; green points are all other odontocete species. Numbers in brackets show the percentage variance explained for each PC.

Figure 4 Cochleae of taxa near the extremes of the morphospace shown in Fig. 3 (not to scale).

(A) Mesoplodon mirus (PC1 max); (B) Globicephala melas (PC1 min); C: Sousa teuszii (PC2 max); (D) Tasmacetus shepherdi (PC2 min).

Figure 5 Principal components (PC) plots for linear measurements for PCs 1-4.

Purple points and convex hulls are Phocoena phocoena specimens; green points are all other odontocete species. Numbers in brackets show the percentage variance explained for each PC.

Quantifying intraspecific variation in Phocoena phocoena

The disparity of P. phocoena in the 3D shape data (sum of variances = 0.014, median centroid distance = 0.100) is lower than in the linear measurement data (sum of variances = 1.487, median centroid distance = 1.091), although the morphospace occupied is larger compared with other odontocetes in the 3D shape data (Figs. 3 and 5). Values of CV vary across the linear measurements (Table 1) but all are below 30% and only area of fenestra cochlearis (FC) exceeds 20%. This reflects low levels of intraspecific variation in cochleae of P. phocoena. Values of CV for all measurements except cochlear volume, axial pitch and ITD are higher than those found in previous studies (Table 1; Table S6).

Table 1 Coefficients of variation (CV) of 12 linear measurements for 18 Phocoena phocoena specimens, along with the mean CV of these measurements from previous odontocete studies (see Table S3 for raw data), the number of studies featuring each measurement (N), and the maximum number of individuals in each of these previous studies.

SSL, length of secondary spiral lamina; W2, Width of basal turn perpendicular to cochlear width; ITD, Inter-turn distance, maximum distance between turns; FC, area of fenestra cochlearis (FC). No previous studies measured intraspecific variation in SSL or extent of SSL.

	This study	Previous studies	
Measurement	CV (%)	Mean CV (%)	N	maximum number of individuals	
cochlea height	7.783	6.793	8	12	
cochlea width	4.842	3.364	4	12	
number of turns of the cochlear canal	3.394	2.478	7	5	
cochlear volume	8.015	10.23	1	2	
cochlear canal length	6.189	3.401	4	12	
SSL	6.619	–	–	–	
basal ratio	8.846	0.000	1	2	
axial pitch	8.688	8.736	5	4	
extent of SSL	3.089	–	–	–	
W2	7.612	2.544	2	5	
ITD	11.93	12.622	2	5	
FC	26.80	9.838	2	5	

Comparing intraspecific variation and interspecific variation

There was significantly more variation among odontocete species than within the specimens of P. phocoena (Table 2; Figs. 3 and 5) when the entire shape of the cochlea was considered, whether we used Procrustes coordinates or PCs from the 3D shape data or linear measurements.

Table 2 Results of Procrustes MANOVA and standard MANOVA analyses using either Procrustes aligned coordinates or principal components (PCs) accounting for 95% of the variance as the response variable, and whether a specimen was Phocoena phocoena or another odontocete species as the explanatory variable.

3D shape data analyses used PCs 1 to 26; linear measurements analyses used PCs 1 to 6.

3D shape data	
Test	df	F/approx F	Pillai	p	
Procrustes ANOVA	1,67	6.005	NA	0.001	
MANOVA	1,67	8.623	0.842	<0.001	
Linear measurements	
MANOVA	1,67	9.090	0.468	<0.001	

When we investigated individual PC axes (Table S7), for the 3D shape data we found that only PC1 and PC3 of the 26 PCs significantly differed between P. phocoena and other odontocetes, and for linear measurements only PC1, PC2 and PC4 (only PC1 and PC2 are significant after Bonferroni correction) showed significant differences among the two groups.

Figures S1 and S2 show how species are spread across PC1/PC2 morphospace according to taxonomic family, habitat, diet, feeding mode, dive type and hearing type. All of these variables except feeding mode were also significantly correlated with cochlear shape (Table S8).

Sensitivity analyses

In our sensitivity analyses, MANOVAs for 3D shape data remained significant provided at least four specimens of P. phocoena were included in the analyses (Figs. S3 and S4).

When the specimens of P. phocoena were compared to other Phocoena species only, there was still significantly less variation within the specimens of P. phocoena using Procrustes coordinates from the 3D shape data (but not for PCs) and linear measurements (Table S9).

Variation in cochleae of P. phocoena was not significantly correlated with the side of the head the cochlea came from, the sex, or skull length of the specimens for either 3D shape data or linear measurements (Table S10). Origin (UK vs. USA) was not significantly correlated with variation in cochleae of P. phocoena for the linear measurements, but was significantly correlated with variation in the 3D shape data (p = 0.038; after Bonferroni correction p = 0.152). This is driven by the large negative value of PC1 for the USA specimen. To ensure this did not influence our intraspecific variation results (see above) we repeated our 3D shape analyses (including the GPA and PCA steps) removing this specimen (NHMUK_1873.6.3.45). The results are qualitatively identical to those including the USA specimen (Table S11).

Discussion

In this study, we tested whether intraspecific variation is a major confounding factor in cross-species analyses by comparing variation within the cochleae of the harbour porpoise (Phocoena phocoena) with those across a broad sample of other odontocete species. Levels of intraspecific variation in cochleae of P. phocoena were lower than the levels of variation seen across other odontocetes. P. phocoena showed higher values of coefficient of variation (CV) overall (ranging from 3.394–26.80%; Table 1) compared to other odontocetes (ranging from 0–23.33%; Table S6), however, for most cochlear measurements the values of CV were similar. The greatest difference in values of CV was for the fenestra cochlearis, which may be artificially variable due to a lack of standardised protocol for measuring it (Racicot et al., 2016; Churchill et al., 2016). Intraspecific variation in P. phocoena might be expected to be higher than in other odontocetes because of their large geographic range which stretches across almost all shallow cool temperate to subpolar waters of the northern hemisphere, within which they encounter many different kinds of habitats (Jefferson, 2014). This suggests that P. phocoena should generally have a higher level of intraspecific variation compared to other odontocetes with more limited geographic ranges and habitats. We cannot test this here, due to the paucity of studies containing more than one specimen for each species, but future research should compare the values of CV found for P. phocoena here with those of another similarly widespread species, for example Tursiops truncatus (common bottlenose dolphin) or Delphinus delphis (short-beaked common dolphin).

Although there was significantly more variation among odontocete species than among the specimens of P. phocoena, for at least some principal components, specimens from other species overlapped with P. phocoena. Cochlear morphology was also related to habitat, diet, mode of echolocation, and dive type (Table S8), suggesting that species identity alone is not enough to explain variation across all principal components. In addition, not all of the specimens of P. phocoena cluster together tightly (Fig. 3). This could reflect differences in cochlear morphology among populations (although this does not appear to be the case for these data as only one specimen is from the USA and the rest are from the UK), or variability within the species. Cochleae also exhibit convergent evolution (Park et al., 2019), so individuals or species that live in similar acoustic environments (regions where variables such as water temperature, pressure, and salinity result in sound travelling through the water at similar velocities) as P. phocoena may have similar cochleae. The relatively low levels of intraspecific variation found in P. phocoena adds further support for the acoustic environment being a constraint on cochlear morphology. This is additionally demonstrated in the morphospace plot of PC1 and PC2 (Fig. 3), the area occupied by P. phocoena is shared with taxa such as P. sinus (vaquita), Cephalorhynchus commersonii (Commerson’s dolphin), and Inia geoffrensis (Amazon river dolphin), all of which also live in shallow waters. However, other taxa that spend more time in deeper waters, for example Pseudorca crassidens (false killer whale) and Tursiops truncatus, also occupy this region of morphospace, indicating that additional factors are also in play, although they may also frequent shallower waters or have coastal ecotypes (Jefferson, Webber & Pitman, 2015). Previous studies have also suggested that constraints on cochlear morphology are strong, due to several factors including frequency propagation (Manoussaki et al., 2008), spatial constraints (Pietsch et al., 2017), and its vital importance for survival in odontocetes (Ketten, 1992a). As such the capacity for elaboration or innovation in cochlear shape may be limited, which could force all odontocete cochleae to fall within the same limited morphospace. Other sources of error, such as mislabelling of specimens in collections or errors in µCT scanning or landmarking, are unlikely due to the care taken during these stages, and assistance from expert curators in museum collections.

Although, as described above, the specimens of P. phocoena did not cluster together closely in our PC plots, we note that PC plots only show two dimensions of the data, not all of the 26 PC axes that comprised 95% of the variance in cochlear shape. Our MANOVA results, however, showed that there was significantly more variation in cochlear shape among odontocete species than within P. phocoena specimens. This highlights that clustering (or not) in several PC axes, does not necessarily equate to clustering across the entire morphospace, although limitations in how we visualise more than three dimensions mean that PC plots are still valuable visualisation tools. In addition, although PCAs on different datasets are not directly comparable, shape variation in the cochleae of odontocetes appears to be spread across a much higher number of PCs than that seen in most studies involving skulls (e.g., Cardini & Polly, 2013; McCurry et al., 2017; Page & Cooper, 2017; Randau, Sanfelice & Goswami, 2019).

Conclusion

The 18 specimens of P. phocoena used in this study represent the highest number of records for one species used in intraspecific studies of cochlear morphology (the next highest is 12 individual sperm whales; Schnitzler et al., 2017) in cetaceans. Overall, our results show that intraspecific variation of cochlea shape in P. phocoena is lower than interspecific variation of cochlea shape across odontocetes. Assuming that intraspecific variation of P. phocoena is representative of patterns across odontocetes, this suggests we can still be reasonably confident in our conclusions when using n = 1 for comparative studies of cochlea morphology, as done in most previous studies (Ekdale & Racicot, 2015; Ekdale, 2016; Park, Fitzgerald & Evans, 2016; Park et al., 2017a; Park et al., 2017b). It is possible that P. phocoena is not representative of patterns across odontocetes. If P. phocoena is less variable than other odontocetes this would reduce our confidence that using just one specimen for comparative studies is sufficient. However, as discussed above, we believe this is unlikely because P. phocoena has a broad geographic range, and shows variability in habitat use and trophic ecology across that range (Bjørge, 2003; Jefferson, 2014). As such we speculate that P. phocoena is instead likely to be more variable than most species making our conclusions more conservative. Ideally, future studies should use multiple cochleae for each species, but where this is impossible due to the rarity of specimens, or expense of specimen loans and/or µCT scanning, this should not influence the conclusions of comparative studies, assuming results from P.phocoena can be generalised across odontocetes.

Supplemental Information

Supplemental Information 1 Ecological data on each species and a reference list for where these data came from

Table S4-Ecological data for each species of odontocete

Table S5-References for ecological data from Table S4

Click here for additional data file.

Supplemental Information 2 Supplemental Information from: Intraspecific variation in the cochleae of harbour porpoises (Phocoena phocoena) and its implications for comparative studies across odontocetes.

Table S1. Specimen accession numbers and museum of origin. AMNH = American Museum of Natural History, New York, USA; IRSNB = Royal Belgian Institute of Natural Sciences; QMJM = Queensland Museum; NHMUK = Natural History Museum, London; NMVC = Museum Victoria, Melbourne, Australia; NMB = Naturhistorisches Museum, Basel, Switzerland. Side refers to the side of the head the cochlea came from.

Table S2. Scan Parameters of newly scanned specimens of Phocoena phocoena at the NHM. NHMUK = Natural History Museum, London. All specimens used a scan power (kV) of 100 and 1,999 slices. Details for all other specimens can be found in (Park et al., 2019).

Table S3. Additional information on Phocoena phocoena specimens. Side refers to the side of the head the cochlea came from. CBL is condylobasal length (skull length) recorded in mm. Note that not all cochleae had a corresponding skull in the collections

Table S6. Summary of coefficients of variation (CV) found in other studies of odontocete cochlea. Where these studies contained right and left cochleae from one individual we first took the mean for the individual, then calculated CV across individuals.W2, Width of basal turn perpendicular to cochlear width; ITD, Inter-turn distance, maximum distance between turns; FC, area of fenestra cochlearis. * This value is based on measurements from right and left cochlea of 12 individuals (i.e., 24 cochlea) rather than the mean value for each individual because only overall means are published in the paper.

Table S7: Results of ANOVAs using principal components (PCs) accounting for 95% of the variance as response variables, and whether a specimen was Phocoena phocoena or another odontocete species as the explanatory variable. 3D shape data analyses used PCs 1 to 26; linear measurements analyses used PCs 1 to 6. Significant p values (p < 0.05) are in bold.

Table S8. Results of MANOVA analyses using principal components (PCs 1 to 26) that account for 95% of the variance in 3D shape as the response variable, and whether a specimen was Phocoena phocoena or another odontocete species (group), and additional taxonomic or ecological variables (various) as the explanatory variables.

Table S9. Results of Procrustes MANOVA and standard MANOVA analyses using either Procrustes aligned coordinates or principal components (PCs) accounting for 95% of the variance as the response variable, and whether a specimen was Phocoena phocoena or another phocoenid species as the explanatory variable. 3D shape data analyses used PCs 1 to 14; linear measurements analyses used PCs 1 to 7. Significant p values (< 0.05) are in bold.

Table S10. Possible correlates of intraspecific variation in the cochleae of Phocoena phocoena specimens. Results are from MANOVAs using principal components (PCs) accounting for 95% of the variance in the cochleae shape as the response variable, and either the side of the head the cochlea came from, the sex, origin (UK or USA), or log skull length (condylobasal length in mm) of the specimens, as the predictor variables. 3D shape data analyses used PCs 1 to 16; linear measurements analyses used PCs 1 to 7.

Table S11. Results of Procrustes MANOVA and standard MANOVA analyses using either Procrustes aligned coordinates or principal components (PCs) accounting for 95% of the variance as the response variable, and whether a specimen was Phocoena phocoena or another odontocete species as the explanatory variable, but excluding the USA P.phocoena specimen (NHMUK_1873.6.3.45). 3D shape data analyses used PCs 1 to 26; linear measurements analyses used PCs 1 to 6. Significant p values (< 0.05) are in bold.

Figure S1: Principal components plots of PC1 and PC2 for 3D cochlea shape data, coloured according to taxonomic family, habitat or diet. Triangles are Phocoena phocoena specimens; circles are all other odontocete species. numbers in brackets show the percentage variance explained for each PC.

Figure S2: Principal components plots of PC1 and PC2 for 3D cochlea shape data, coloured according to taxonomic feeding mode, dive type or hearing type. Triangles are Phocoena phocoena specimens; circles are all other odontocete species. numbers in brackets show the percentage variance explained for each PC.

Figure S3: Median p values from multivariate analyses of variance (MANOVA) on principal components (PCs) of 3D cochlea shape data with varying numbers of Phocoena phocoena specimens included. For each number of Phocoena phocoena specimens we ran the analyses for each possible combination of specimens. Significance at p < 0.05 is indicated by the red dashed line.

Figure S4: Kernal density plots showing p values from multivariate analyses of variance (MANOVA) on principal components (PCs) of 3D cochlea shape data with varying numbers of Phocoena phocoena specimens included. For each number of Phocoena phocoena specimens we ran the analyses for each possible combination of specimens. Significance at p < 0.05 is indicated by the red dashed line, which is not visible for n > 9.

Click here for additional data file.

We thank: Oliver Lambert (IRNSB), Gabriel Aguirre Fernandez (UZH), Brett Clark and Vincent Fernandez (NHM), Alistair Evans (MU) and Loïc Costeur (UB) for scanning several of the specimens in this study; Richard Sabin and Roberto Portela Miguez (NHM), Erich Fitzgerald and Karen Roberts (NMV), Neil Duncan and Eileen Westwig (AMNH) for access to specimens.

Additional Information and Declarations

Competing Interests

Author Contributions

Data Availability

The authors declare there are no competing interests.

Maria Clara Iruzun Martins performed the experiments, analyzed the data, prepared figures and/or tables, authored or reviewed drafts of the paper, and approved the final draft.

Travis Park and Natalie Cooper conceived and designed the experiments, performed the experiments, analyzed the data, prepared figures and/or tables, authored or reviewed drafts of the paper, and approved the final draft.

Rachel Racicot conceived and designed the experiments, authored or reviewed drafts of the paper, and approved the final draft.

The following information was supplied regarding data availability:

The specimen accession numbers for P. phocoena are available in Table S1; the locations and further information are available in Table S3.

P. phocoena data are available at Martins, M.C.I., Park, T. and Cooper, N., 2019. Dataset: Intraspecific variation in harbour porpoise cochleae. Natural History Museum Data Portal (data.nhm.ac.uk).

https://doi.org/10.5519/0091362.

The specimen accession numbers of other odontocetes are available in Table S1 of Park, T., Mennecart, B., Costeur, L., Grohé, C. and Cooper, N., 2019. Convergent evolution in toothed whale cochleae. BMC Evolutionary Biology, 9: 195. https://doi.org/10.1186/s12862-019-1525-x.

The other odontocete data can be accessed at Park, T., Mennecart, B., Costeur, L., Grohé, C. and Cooper, N., 2018. Dataset: Convergent evolution in toothed whale cochleae. Natural History Museum Data Portal (data.nhm.ac.uk). https://doi.org/10.5519/0082968.

The code is available at Cooper, N., Martins, M.C.I., and Park, T., 2020. GitHub: nhcooper123/intraspecific-porpoise: code for the paper. Zenodo. DOI: 10.5281/zenodo.3693950.

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
