# Peer review of "Intraspecific variation in the cochleae of harbour porpoises (Phocoena phocoena) and its implications for comparative studies across odontocetes"

_PeerJ, doi:10.7717/peerj.8916_

## Round 0.1 · original submission · Major Revisions

Thank you for your well written and interesting study. The biggest question raised by multiple reviewers, and I agree with them, is the definition of interspecific variation. I believe more context (age, population, sex) would help with the overall presentation. Otherwise, the issues are minor and can be easily addressed.

Reviewer 1 ·

Basic reporting

The article is written in a clear, concise manner that is easy to read and easy to follow. It uses professional and unambiguous english throughout. The references are complete and thorough, and no major references are missing. The data has all be shared and is accessible, and the figures and tables are complete. The article is self contained with clear hypotheses being tested and results that unambiguously shed light on those hypotheses.

My only suggestion would be pertaining to the data. The authors have made most of their raw data available NHM dataportal. This includes all of the landmark data and the landmarks themselves. However, I would really like to see the authors publish the actual 3D files of each cochlea, probably as an STL or a PLY. I understand that this data also appears in a Park et al MS that is in review, so I realize they may be unwilling to do this at this time. I do not think this should stop the MS from moving forward, but I do think the 3D files should be made available.

Experimental design

The others lay out a clear and well stated hypothesis and the experimental design explicitly tests it in a falsifiable manner. It is clear how the results address the hypothesis and it is clear why the results are relevant to the field at large (i.e. many other studies are limited to a single specimen). The methods section in particular is exceptionally well written, easy to follow, and easy to replicate, and should represent a gold standard for methods reporting.

Validity of the findings

I have no concerns as to the validity of their findings. Their conclusions logically follow from their results.

Additional comments

A few additional edits and comments for the others to consider during revisions:

Line 46: I agree with the overall point, but it is probably worth adding a small caveat. Using a single specimen as representative of a species shouldn’t be a problem assuming that all species show similar levels of intraspecific variation as harbor porpoises. Perhaps a simple “if the results are consistent across other species…”

Line 60: “this has not been extensively tested”. Do you mean for cetaceans specifically? For cetacean cochlear data? I would argue that it has been extensively tested in some taxonomic groups for some traits, but I would agree that it certainly hasn’t been extensively tested for cetacean cochlear morphology.

Lines 205–212: Generally, I think that plotting PC1 and PC2 with color to indicate the ecological variables is a good idea. I suspect the authors and I would disagree politely over some of the categorizations—specifically pertaining to what may or may not constitute a suction feeder—but I think the plots are interesting and informative and, especially as supplementary, not something I would scrutinize too closely. I would, however, like to see this broken down, perhaps in another supplementary table? For example, in figure S6, which taxa are the ones considered “raptorial/suction” or which taxa are considered “deep”. As a reader I would appreciate knowing which taxa belong to which category and the references on which that is based.

General: The authors primarily compare harbor porpoises to the rest of Odontoceti as a whole. However, I am curious whether the dataset allows you to compare (even in a limited manner) harbor porpoises to other species of phocoena. I noticed a few in the dataset, and I am curious if they specifically are closer in morphospace to the P. phocoena dataset? This could be relevant because it might indicate that, intraspecific variation is more impactful if the taxonomic scope of the study is narrow. I don’t think the authors need to do much more, just perhaps add a sentence or two about this to the discussion.

Reviewer 2 ·

Basic reporting

I cannot agree that the manuscript is written in a clear, unambiguous, professional manner. The coverage of literature is sufficient, but background lacks specific, focused insights, and contexts are unclear (the authors never specified what intraspecific variations are examined in the cochleae in the first place). The manuscript is structured professionally, except that supplementary figures and tables should be incorporated into the main article. The manuscript is self-contained, but there is no statement of a specific hypothesis.

See general comments for what specifically the manuscript failed to meet these standards in, and how to improve.

Experimental design

The manuscript is original, primary research, but did not define research questions (intraspecific variations as a confounding factor in taxonomic identifications or comparative analyses, as a correlate of echolocation, or... The narrative is vague and waddles between these different questions). I am not sure what knowledge gap the manuscript fills. Technical and ethical standards are upheld, and methods were described with sufficient details.

See general comments for what specifically the manuscript failed to meet these standards in, and how to improve.

Validity of the findings

The authors surely quantified variations in cochlear morphology, but they never put it in context (sex, age, size, population, etc.). Any trait varies within a species, but the key information is its correlates. The authors tucked this most relevant aspect of their analysis in supplementary information (where they looked at vaguely defined ecological traits), and even this was only interspecific and did not extend onto their discussion of intraspecific variation.

They are unclear about which data specifically derived from Park et al. (2019) or Martins et al (2019). It almost reads like a lot of data got transferred over from these papers, but they didn't detail which specimens, species, or measurements. It is not clear if 3D renderings are available in NHM Data Portal.

Conclusion is not clear. Implications are not clearly stated or discussed.

See general comments for further details on what specifically the manuscript failed to meet these standards in, and how to improve.

Additional comments

I carefully read “Intraspecific variation in harbor porpoise (Phocoena phocoena) cochleae and its implications for comparative studies across odontocetes.” In this manuscript, Martins and colleagues report results from geometric and linear morphometrics on the cochlea from harbor porpoise (n=18) and 51 other odontocete species. Intraspecific variation (as ambiguously defined by the authors) was smaller substantially than interspecific variation (also as ambiguously defined by the authors). I found the methods and results are overall competent, but the presentation requires massive improvement. My concerns fall into four major categories: 1) treatment of intraspecific variation; 2) tenuous rationales and overgeneralizations, and the lack of the sense of goals; 3) data presentation; 4) overall writing; and 5) specific technical comments.

1) Treatment of intraspecific variation
The authors should consider what “intraspecific variation” they are recovering in the cochleae of harbor porpoises. Intraspecific variation —any polymorphism in a species, however defined— is so broad that it means little without context. Sexual dimorphism, ontogenetic traits, allometric trends, developmental plasticities, population means, reaction norms, performance differences…. They are all intraspecific variations. Many of these contexts (such as sex, size, age estimates) are essential collections data and I assume they were available to the authors. At the very least, size can be easily taken from their own scans. There should be some effort to dissect what biological context the data speak to.

In terms of study design, I also wonder why they only picked one specimen from each of the other species. It is unlikely that only one each was available to the study. I wonder if they picked one from each for the sake of simplicity. I believe they should sample whatever small numbers available in at least two or three more species, and they could test whether the range of variations in P. phocoena is compatible with other odontocetes — which is potentially an important point they could have backed up with data.

Additionally, what collection bias might exist in those other odontocete species sampled for this study?

2) Rationales and generalizations
The authors should have a clear sense of goals, or what they want to accomplish in this manuscript. The lack of this understanding transpires through unclear rationales. The authors were unclear about how and why the harbor porpoise data inform us broadly about morphological variation of cochlear among and within odontocetes — only a vague, arm-waving explanation how common they are in the collections and how they distribute in wild. The authors argue that intraspecific variation may be a confounding factor. But they omit explaining how specifically (e.g., are cochlear characters frequently used for specific diagnosis of odontocetes, and, if so, what?). They do not seem to refer to parallel studies in other taxonomic groups they only cited in passing, such as Cerio & Witmer, and didn’t seem to incorporate the analytical pipeline or interpretations from such previous studies in their own manuscript. The authors quantified intraspecifc variation to understand how it is set in the odontocete background, but they did not care to consider or test for sexual dimorphism, ontogenetic variations, allometric trends, or left-right asymmetries. Overall, it gives an impression that the authors employed a wobbly narrative.

Related to this problem, they make many general comments, either out of context or without sufficient evidence. Examples:
- Intraspecific variation as “key driver” of evolution by natural selection. (Line 50) Do they really mean a “driver”?
- “There can be variation in survival defences (what is “survival defence”?), parasite resistance, resource manipulation and others, all of which can alter the number and strength of interactions shifting the dynamics of an ecosystem.” (Line 53-54) What do the authors want to accomplish here? The list is neither comprehensive nor pertinent to their discussion.
- The authors’ key statement: Levels of intraspecific variation are lower than those of interspecific variation (Line 59-60). In what context? Intraspecific variations CAN be greater than the sum of means across species.
- Intraspecific variation not considered or impossible to calibrate (Line 62-65). If this concern is really the motivation for the authors, they must justify why harbor porpoises and why cochlear morphology. If the availability and its importance in echolocation are their best rationales, then this motivation about the general nature of intraspecific variation is irrelevant. The authors should identify something like: echolocation is an important ecological trait -> then they should look into how different cochlear morphology correlates with what aspect of echolocation. If taxonomic misidentification is a concern -> then the authors should list examples of diagnostic characters in cochleae.
- The focus of study is “hearing ability” (L 107). Then why no specific discussion on correlation between cochlear morphology and auditory function anywhere in the manuscript?
- Semicircular canals not phylogenetically or ecologically informative (Line 107-108). Really?
- They can be “interpreted biologically” (Line 136). This statement doesn’t make sense. Any biological measurements can be interpreted “biologically.”

The authors should consult other parallel papers (such as Cerio & Witmer, which they cited) for how they set up their analysis, and in what context the data should be analyzed and interpreted. These out-of-context, truistic general statements are not helpful because the authors are not discussing general nature of intraspecific variation at all. They are discussing some specific aspect of intraspecific variation (however defined) in a given structure of a given group.

3) Data presentation
I recommend the authors include as many 3D renderings as possible in main figures. This is: a) to give readers good idea of the cochlear morphology and metric traits (Fig. S1); and b) to give readers sense of variation in addition to PCA plots — especially if they mean to maintain PCA results are “misleading”. Also, PeerJ as an online venue has no physical limit to the number of figures. So there is no good reason not to do this.

I also encourage all supplementary figures and tables incorporated in main text and discussed extensively. Again, no limit on length provides an opportunity to be as comprehensive in your presentation as possible. Few choose to view every single supplementary material. A good way to make sure your data to be open is to include them all as main figures and tables.

4) Overall writing
The authors did competent analyses and produced valid results. This is confounded, however, by the lack of clarity or focus in overall presentation and writing. Please see attached pdf for specific comments. I was distracted by general discussion of intraspecific variation (most of which seems to require no introduction or discussion, and reads like reinventing a wheel; e.g., the first three sentences in Abstract) and the lack of focus in the topics they set out to explore (e.g., a rough sketch of potential ecological correlates in Discussion) or sometimes an entire neglect of it (e.g., no discussion about sex, size, or age correlates). This aspect requires massive improvement by using clear and focused narrative.

Also, some minor corrections:
- Throughout the manuscript, it is not “18 P. phocoena specimens”. It is “18 specimens of P. phocoena.” Latin binomials should not be used as an adjective.
- “As we expected” “As we might have expected” and the like (Line 284, 313, and throughout) — these phrases do not accomplish anything. If you made predictions, then state them specifically and test them.
- Fairly variable (Line 286). What is “fair” variation?
- Line 292 – “coefficient of variation values” should be “values of coefficient of variation.”
- Line 300-306 should be in Introduction to justify the work.
- “Were nested with” (Line 310) — No, simply “overlapped with”.
- “Neatly” (Line 313) — please be more descriptive.
- “Although this does not appear to be the case for these data” (Line 315) — why? I do not understand.

5) Some technical comments

The list of linear measurements included some non-linear traits, such as angles, areas, volumes, discrete counts, and ratios (Line 128-134). Then the authors stated that they log-transformed all metric traits (Line 153). Did they log-transform ratios? (and how for ratios < 1?) Angles? Counts?

Line 157-160 – The authors should clearly state whether 3D renderings are available in NHM Data Portal, and which specific data entries derived from Park et al. (2019) or Martins et al. (2019). All specimens of P. phocoena?

Line 214-225 – the authors discuss some sensitivity analysis, but I don’t see detailed results of this sensitivity analysis. Also, how are those specimens chosen? Strictly randomly or otherwise?

Line 286-290 – the authors argue a standardized protocol could reduce noise. But this shows that the authors had an opportunity to make that sensitivity test but did not bother to do it. Please show that if the standardized protocol could reduce noise.

Line 292-296 — “artificial variation”. If the authors are sure about artifacts, then why show deficient results that they know contain artifacts? What did the authors do to minimize this?

Line 315 “Plasticity” — I think “plasticity” here is taken out of context, and I don’t know what the authors really want to communicate with this term here.
Line 319 “Similar acoustic environments” — what environments are “acoustically similar”?

Line 328-330 “same limited morphospace” — This argument is unclear. The least it suggests is that there is no one dominant correlate that explains most of the shape variation. I'm not convinced that the authors can say any more than that given the data.

Line 330-333 and also elsewhere discussing or mentioning human errors — Why go look for these human errors, where (a) the variations can be reasonably interpreted in biological context and (b) the authors themselves reject these possible sources of error? Do the authors think that could explain some major patterns in data? And if so, why are they presenting this result?

Line 335-350 General discussion of PCA interpretations — I don’t find this entire paragraph helpful. First of all, the authors complain that these plots do not covey “true” results because the plots show distributions that are not perfectly aligned with their “expectations”. But, once again, the plots seem to me as reasonable distributions, and they can be reasonably interpreted. I don’t know what the authors want to accomplish here. Are they suggesting there is some “true result” or “true interpretation” that they can’t visualize otherwise? For the sake of argument, assume that their disappointment has some ground. Then I would wonder why they chose to show those “misleading” plots anyway. I recommend that this whole paragraph be removed.

Line 355-356 “lower than expected” — the authors never showed what was the expectation.

Line 362 “we believe this is unlikely” — Why? “As discussed above” — where?

Line 363 “likely to be more variable than most species making our conclusions more conservative” — I do not follow their argument here. Why can they claim that when they only looked at one from each of the other species?

Annotated reviews are not available for download in order to protect the identity of reviewers who chose to remain anonymous.

·

Basic reporting

Martins et al. present a thorough and interesting examination of morphological variation in the cochlea of harbor porpoises, using the largest sample size of CT scans to date of an individual species of odontocete whale. They identified low amounts of intraspecific variation in the cochlea shape within harbor porpoises, much lower than that seen across Odontoceti as a whole. Although some variability was still inherent, the study suggests that the use of single specimens in comparative studies of inner ear morphology should not present issues in data interpretation.

This study is extremely well written written, and in fact is maybe one of most typo free drafts of first submission I have come across. The study also addresses the important topic of individual variation. Too many paleontologists and marine mammal specialists tend to discount individual variation when describing morphology, and I commend the authors for actually rigorously testing the importance of this aspect of morphology in inner ear data. The paper is well structured, and for the most part well referenced (see below).

Experimental design

The analytical methods and data all seem sound, and the analyses thorough. I don't really have any suggested edits on the methods they used. I do have several concerns with the data, which I will elaborate below.

First, what is the origin of the P. phocoena ears? P. phocoena has a wide distribution, and populations in different parts of their range have a significant degree of morphologic and genetic differences. The case in this paper is much stronger if these all derive from a single geographic region or population. Furthermore, if there are specimens from, say the Pacific and Atlantic, does some of the variation observed reflect those differences in population? This needs to be spelled out more clearly in the paper. I would also suggest the authors state how many specimens are right vs left, given that the paper mentions some evidence of bilateral asymmetry in these structures. Does this account for some of the variation observed?

My second major concern focuses on the interspecific variation, particularly the classification of odontocetes by habitat, diet, and feeding mode. Although definition for some of these classifications, such as diving depth and hearing type are explicitly described, classification of the prior categories is left vague.

There is an extensive degree of variation in habitat use, so what particular parameters are used to define an animal as nearshore and offshore. I will note that the authors seem to be aware of this particular issue, as unmentioned in the text but illustrated in the relevant figure are categories that combine multiple habitat types (for instance nearshore/offshore as a separate category.

Similarly, I assume fish specialists are taxa whose diets consist of more than 50% fish, while generalists have a more equal fish versus squid ratio. But those proportions should be spelled out in the paper.

The same vagueness can be seen in classification of prey capture. Many odontocetes are capable of both raptorial and suction feeding.

In addition to more explicitly spelling out how these animals are characterized, there also needs to be references given here. And if all of these categorizations are simply following classifications laid out in a past paper, that paper should be cited as the source.

Validity of the findings

This is a well put together paper, and other than some tweaks suggested above, I don't doubt the overall findings. Everything is clearly written, structured, and spelled out, and the conclusions seem solid with no real speculation inserted within.

About the only extra comment here I would have follows from my above suggestions. There should be an excel file or word table in the supplementary material which explicitly states all of the classifications for each taxa, ideally with citations. As it currently stands, I don't know where the classifications are coming from nor what whale is under what category, which hampers interpretations of the plots in the supplemental material.

Otherwise however this is great

Additional comments

I only really noticed two sets of typos worth mentioning. First, I would reduce the use of SSL as an acronym, especially as there is one sentence that uses it twice, e.g. "Length of SSL (6) was measured to calculate percentage of extent of SSL (9)...". It would be easier to read if this was just spelled out.

For Figure S6, divetype and hearingtype should be each two separate words, as well as verydeep in the key. Mid should be changed to middle depth or something along those lines. I find it also slightly distracting that the order of the key goes from deep, mid, shallow, verydeep, rather than shallow, mid, deep, verydeep (or the reverse).

Otherwise I didn't notice much in the way of typos or grammatical edits.

Ultimately I really enjoyed reading this paper, and I think the study is excellent. I commend the authors on their work, and I look forward to seeing this published.

Best,

Morgan Churchill

---

## Round 0.2 · Major Revisions

I understand that you have tried to address the reviewer's comment in the previous revision and that your study is sound. It appears to me both reviewers believe there is room for improvement in terms of presentation. Specifically, the detailed feedback of reviewer 2 should better communicate his/her suggestions with regards to creating an updated figure to illustrate the variations in morphology, discussion on intraspecific variations, and clarification of the reconstruction techniques. Hopefully, you will find them as reasonable and achievable edits to make.

Reviewer 2 ·

Basic reporting

I still cannot agree that this manuscript passes the standard for clear, unambiguous, professional language. My comments follow. Please see the referee report.

Experimental design

Nothing to add.

Validity of the findings

Please see the referee report.

Additional comments

I carefully read the revised manuscript “Intraspecific variation in harbor porpoise (Phocoena phocoena) cochleae and its implications for comparative studies across odontocetes.” Firstly, I appreciate many changes the authors made in response to various comments from the referees. Secondly, however, I was disappointed to see rhetorical devices the authors set up in the revised text and their rebuttal to avoid addressing fundamental issues with the paper. This is puzzling — They did a competent analysis, but continue to refuse putting their findings in a context.

Example: A recent paper by the same team of authors “Convergent evolution in toothed whale cochleae” (BMC Evolutionary Biology 19, 195) states the following conclusion: “The extreme acoustic environment of the deep ocean likely constrains cochlear shape, causing the cochlear morphology of sperm and beaked whales to converge. This study adds support for cochlear morphology being used to predict the ecology of extinct cetaceans.”
This manuscript can build on this paper by discussing how “low” levels of intraspecific variation supplements the ecological constraint argument, predicting how their findings can be generalized within odontocetes based on the ecology of harbor porpoises, and even providing some counterpoint: the fact that variations within harbor porpoises can still overlap with some mainly deep-water species (L 388–398) suggests there is more to it than ecology. What more? Do regions of the cochlea more variable within harbor porpoises match regions showing ecologically constrained convergent evolution elsewhere in odontocetes? Or do they differ?
Instead, the authors touch on Park et al. (2019) by simply mentioning the intraspecific variations in harbor porpoises overlap with some shallow-water species and some mainly deep-water species. No further discussion is provided (L 388–398).

Now, I would normally keep it to the manuscript and do not cite rebuttal by authors to argue my points, but in this case it is necessary as the authors chose to refute suggestions rhetorically, not scientifically.

For example, they refuted suggestions of including images of cochleae or supplementary figures mentioned in the text as main figures: “Other reviewers and the editor have not suggested moving supplementary figures and tables” or “People don’t need to see every single cochleae image to understand the results of the paper. If they really want to look at them they are available in our online data.”
Here is my point: How does a paper with “intraspecific (morphological) variation of cochleae” in the title obstinately refuse to illustrate the very variations (or subtlety thereof) picking from many examples in the reconstructions they have — let alone showing a single image of the structure they study in the main figure? They could do this by showing two reconstructions, one each from male and female, or young and adult, or any demographic extremes, to give the readers an idea how a cochlea looks like, and what levels of morphological variations they are dealing with. They could do this by showing extreme shapes at each end of the PC axes. They could do this by showing examples of cochleae from different ecological guilds of odontocetes alongside one from a harbor porpoise to show interspecific variations far greater than intraspecific. They could do this in a separate figure, or add them in the current figure. They could even simply move one supplementary figure to the main paper to show landmarks. They have tons of data in reconstructions they made, and it seems as just a matter of pulling images from the portal they listed in the manuscript.
They did none of this, and simply suggest go look into supplementary files. If they are genuinely interested in communicating their results with colleagues who do not look at dolphin cochleae all the time, they should not resort to such sloppy excuses regardless of what other referees endorsed uncritically.

This is unfortunate. The analytical results are defensible, but the authors keep refusing to make seemingly easily attainable improvements that could help the paper become more accessible and useful. So I continue to argue that this paper needs substantial revision to meet the presentation standard.

Rebuttal letter by the authors
Again, this is highly irregular to comment specifically on a rebuttal letter. However, the authors tried in the letter to justify changes that they did not make. So I will comment on some aspects of the letter to urge them consider their revision efforts seriously.

“The comment “any polymorphism in a species, however defined— is so broad that it means little without context” could easily be applied to interspecific variation given that interspecific variation also includes many sources of variation that are not identified.”
— A straw man argument. While they are at it, interspecific variations are, however defined, systematically meaningful because of its context in interspecies. Intraspecific variations may be discrete variations among populations, sexes, habitats, or any biological factors that the species interacts with, and its systematic significance is unclear until putting it in any of those contexts. So, again, if their purpose is to serve future taxonomists, there are many things I suggested initially and in this round that help them address that. If their purpose is to delineate interactions between echolocation and morphology, there are things they can do to make a clear statement to that effect. They are doing neither.

“Intraspecific variation can meaningfully be defined as variation between species that are not species differences.”
— I have no idea what they mean by this. Intraspecific variations are simply any variations that are not accounted for by species difference.

“We are focussing on the odontocete context as the reviewer notes. It therefore was outside the scope of the paper to cover intraspecific variation across all other species too.”
— But they do discuss how general their findings may be across tetrapods in the manuscript. (e.g., L 438-440)

“Here we are describing intraspecific variation as a general concept, to clarify why it is important in populations and why it is present. In the line the reviewer is querying we are referring to work by Bolnick and colleagues.”
— Why don’t they list examples more relevant to cochlea and its function? How relevant are predator defense or parasite resistance to an inner ear?

“The motivation of the study and why harbour porpoises and cochlear morphology was used is mentioned repeatedly throughout the study. As such we don’t see any justification for completely changing the framing of the paper, especially when reviewers 1 and 3 praised the writing and clarity of the original manuscript. ”
— The authors argued “intraspecific variations are important to characterize” and “cochleae serve important functions” but never connected the two, for which they need some context. By the same logic, they could very well have looked at intraspecific variations in the entire endocasts as they scanned the skulls. Would that be more important? I suggested, as in my comment previously, to bring in examples of diagnostic characters in the cochlea, or link echolocation to specific aspects of variations in the cochlear shape/size. It does not “completely change the framing of the paper.” It’s a clarification that helps their paper.

“We asked the editor about this and they agree these should stay in the supplemental. We think maintaining a narrative in a paper is more important than bombarding readers with all possible figures.”
— I made the recommendation precisely so they could maintain the narrative. The narrative read rather broken and confusing, and it was even more disruptive to drop where one was reading and go look for that supplementary figure or table. They referred to the materials in the main text as supporting evidence, not just raw or giant data tables. Why including them would break up their narrative?

“3D cochlear models can be constructed in different ways and using different techniques. The shape of the fenestra cochlearis varies between taxa, with some having a more circular opening and others having a more elliptical opening. This study and previous ones have used the formulae to calculate the area of these shapes as approximations of the area of the fenestra cochlearis. However, the fenestra cochlearis of each taxon must therefore be designated as one of these two shapes. This means that taxa with a fenestra cochlearis that isn’t exactly the same as its designated shape will have an incorrectly estimated area value.”
— I think this is precisely the information they should be discussing instead of making a hand-waving mention about “artificial variation”. They can paste this response right into the main text. Why not do that?

“This is the discussion, these statements are speculative and based on interpreting our results along with other research on cetacean cochleae.”
“Again this is a discussion, where it seems pertinent to lay out possible caveats and limitations of the analyses and results. Most discussions contain this kind of information without a requirement to fix the whole field.”
— Sure, but those speculations should be realistic, practical, and citable. This is where they could have included the information like their previous comment about reconstruction techniques, which clearly delineate potential sources of such ‘human errors’. I am rather puzzled why they did not.

“We have changed the word “true” to “complete” to allay some of these concerns. However, we feel this is an important caveat of the methods we are using.”
— It’s still there (L 410).

“Often a huge amount of emphasis is placed on clusters visible in plots of PCs 1 and 2, but for highly multivariate data, like our 3D analyses here, PC1 may not actually reflect much of the variation in the data. Why do we still show these plots? They are standard for the field so omitting them would be strange. If another way to represent multivariate data in more than two/three dimensions on a page we would be delighted to use this, but the human brain, and print media, are limited to low dimensionality representations.”
— Again, they could have inserted this response to the text and cite examples they found. To me, these caveats are simply common sense, and common sense requires no repeat unless there are cases of violations. If they can provide citations where these concerns directly apply, it’s worth discussing. But if everyone follows green light, is it worth telling them to not ignore red lights?

“The end section of the discussion, like most discussions, is speculative. We are saying that theory suggests that P.phocoena should be more variable than other species with narrower ranges and niches.”
— And I am saying it does not logically follow, unless the authors can link variations with geographic ranges. Why should one expect A follows B? There are many “theories” that could support this “assumption” of theirs: population genetics, heterogeneity in habitat preferences across the range, patterns of variations in other aspects of anatomy, etc., relative to the generally defined odontocete background.

Comments on the main text (major)

I will first list technical/conceptual comments to which I call the editor’s attention.

L23–29
This is casting way too broadly. It belongs in Introduction.
Also, low intraspecific variations do NOT necessarily mean "studies can proceed with only a few individuals." Small variations can still be morphologically discrete, taxonomically significant, but seemingly "low" when only overall shapes are considered -- consider bone textures, sutural forms, histological features, presence/absence of carinae, denticles, or other minor features, and many others.
My point here is that this is a wrong extrapolation, and this type of information that the authors can't substantiate doesn't belong in Abstract. Delete L23-27. Replace “Meaning that studies can proceed…” with “but this assumption is not usually tested.”
In any way, this narrative doesn’t work unless cochlear characters are cited as diagnostic in odontocete systematics.

L49 Introduction
This is supposed to provide context and justification: why cochleae? Why not, for example, endocasts?

L64-65
At whom are the authors pointing fingers? Do they really think intraspecific variations are not "considered at all"? Most taxonomic diagnosis is written with a list of traits, with considerations about intraspecific variations (e.g., those known to be variable intraspecifically in relatives are usually carefully omitted). Just because they don't use the term in text does not mean they didn't consider intraspecific variation.

L97-98
The authors in their rebuttal letter state that there are taxonomically diagnostic characters in the cochleae. What are those traits, and can their geometric methods capture those traits? This is an important point to justify their study, but I still do not see it. All they have to do is to cite some examples from odontocete descriptions.
In addition, why do they not discuss in details what they themselves have done with odontocete cochleae recently?

L102 Data collection
It appears that the authors have used the same data source as other studies (Park et al. 2019). They could make their life easier by explicitly stating this. Why downplay it?

L113
Why do the authors not say they used the same data?

L179-182
So what did they took? I don't understand what "cochleae data" mean. Reconstructions on which their measurements and landmark data are based exist in the data portal as Martins et al. (2019). Linear measurements and landmark data also came from this dataset, or got transferred from Park et al. (2018 or 2019)? Then what data did the authors actually collect SPECIFICALLY and NEW for this study? It is entirely fair to say they used the preexisting dataset, rather than to be vague about what data they collected firsthand for this study.
And I will repeat this. DO NOT use Latin binomial as an adjective. It's not P. phocoena cochleae. It's cochleae of P. phocona.

L264-268
But how can they test and rule out left-right asymmetry when there are only three specimens represented by the right cochlea against 15 from the left side?

L361 “may reduce variation…”
Or increase? It is also possible that the reconstruction methods somehow underestimates whatever 'true' variations the authors seem to be envisioning. Is a standardized protocol actually better at capturing morphology? or simply time-saving? If you assume intraspecific variations in bone densities, or collection and curation somehow affecting the quality of the specimens, or natural standing variations affected by any other factors, for example, a standardized protocol could actually increase noise.
I really think the authors should focus on what is practical, and what are realistic improvements they might expect. They do not have to create the narrative of false positives (or negatives).
Also: Is large variation bad? They seem to imply it in this paragraph.

L375-376
values of CV, or relative degrees or magnitudes of variations more generally? CV values are dependent on the dataset, so won't those be incomparable, strictly speaking?
L409-411
No. This is a given. If they are not communicating the 'true results of multivariate analyses" clearly in this paper, then it would defeat the purpose of the entire paper. Why do they undermine themselves like that?
I think they are trying to say something, but are using wrong words to describe it and I cannot guess what it is instead.

L413-414
It's given for PCA. Again, they are basically implying here that there is more to their analysis than they show in Fig. 1. If they cannot communicate exactly what, then they are not communicating their interpretations of their own results clearly.

L415-421
To what choir are they preaching as if they invented the method? If they are to tell colleagues how to do PCA, they should cite where people failed to do this. I cannot think of an example.

L421-424
Once again, PCAs based on different datasets are NOT strictly comparable with each other. Did they sample identical landmarks/traits from an identical set of specimens? This is not a valid argument because they misunderstand PCA. What extra care exactly do the authors argue we need to take?

L431-433
How can they say that based on the study of a single species? Because P. phocoena is a broadly distributed species and expected to interact with different regimes, because it is morphologically conservative in other aspects of anatomy, because population genetics of the species, like other odontocetes, suggest the species has standing genetic variations comparable to other species of odontocetes... Or? They really need to address the assumption they are making to extrapolate P. phocoena onto the general trend among odontocetes.

L435-438
Why do you expect broad geographic ranges to enhance intraspecific variations? Make the logical link clear. This expectation may not be reasonable if the species is homogeneous about habitat usage and proportions of time spent in different habitats. Any evidence supporting that claim? or any other lines of inference can they make (as in my comment above)?
Also, “false optimism” is a poor choice of words.

L438-440
Really, these studies are about relative degrees of intraspecific variation against arbitrary systematic background. The authors here compared morphological variations against odontocetes, but is that really comparable whey other authors might have tested against a genus, family, etc? Are there not others that report measure of intraspecifc variations, and can they not support and draw some general conclusion like "regardless of systematic levels to a certain point, cochleae tend to be morphologically conservative.

L442-443
I asked this before and didn't get the answer: Is there any example of cochlear characters in odontocete diagnoses, and where this came under the discussion of taxonomic distinction? If there is, that would be a useful thing this paper can mention and provide insights for.


Comments on the main text (minor)

I will follow these comments with more specific, grammatical/stylistic corrections.


L25 “In most comparative studies…”
In most morphological traits

L29 “This assumption is yet to be tested…”
Are the authors saying that intraspecific variation of any sort has never been tested in cetaceans? I find that hard to believe.

L 31 “Studies of cochlear variation are common in odontocetes…”
Interspecific variation of cochlear morphology is well characterized among odontocetes

L31-32 “Because of its importance in…”
Because of the importance of the structure in…

L34 “cochlear variation”
Variations in cochlear morphology

L63 “It is assumed…”
For most morphological traits, it is assumed…

L63 “levels”
magnitudes, or degrees

L63 “lower compared to…”
smaller than those of…

L65-66 “or using multiple specimens…”
“this assumption is not (or cannot be) tested either because of logistical constraints”

L66 “more specimens simply do not exist…”
“samples are limited”

L75 “for feeding, communicating and finding a mate”
Is that all, and are these examples even parallel with each other? (finding a mate can be, broadly, communication.)

L78 “due to”
“because of”

L79 “scanning them takes time and money”
“scanning them is logistically unreaslistic”

L81-82 and throughout
Something is wrong with their citation manager. Separate entries of Ekdale and Racicot et al. There are many other examples throughout the text by those and other authors. Fix them please.

L89-91 “comparing variation within the cochleae of 18 specimens of harbour porpoise (Phocoena phocoena) with cochlear variation across 51 other odontocete (toothed whale) species.”
“comparing morphological variation among the cochleae of 18 specimens harbor porpoise…with those across 51 other species of odontocetes (toothed whales).”

L92 and throughout
I repeat this until it sinks in. DO NOT use Latin binomial as an adjective, period. Cetacean cochleae, and cochleae of Phocoena phocoena. The authors repeat this error throughout the text, including in the title. The authors appeared to use "Replace" to make changes for "specimens of P. phocoena" but did not correct others. Search for the binomial and fix throughout.

L115-117 “Of these 15 are from the left-hand side of the skull, and three from the right-hand side. 17 are from the UK Coast, and one is from the North Atlantic, four are female five are male and 11 are unsexed (see Table S3). All specimens are adult animals.”
“Of these 15 are from the left-hand side of the skull, and three from the right-hand side; 17 are from the UK Coast, and one is from the North Atlantic; four are female, five are male, and 11 are unsexed (see Table S3).”

L123 “Figure S1”
What is their justification NOT to put it in the main figure? Puzzling.

L254 “so on”
What on? Until what? What they are describing here is pairwise comparison for GPA, PCA, and MANOVA, followed by every combination of three specimens, and SO ON UNTIL reaching every combination of 17 specimens?

L258 “other Phocoena species”
Either "other species of Phoena" or "other Phocoena spp."

L295 “basal ratio and extent of”
Extent of what?

L332 “speciemns of P. phocoena”
I noticed this elsewhere and am only making a note of this here. I'm not a native English writer but don't they need "the" for this and some (many) of these elsewhere in the text, given context?

L337 “Variation in…”
Also, distinguish variation versus variations. I think it should be plural here. “Morphological variations in the cochleae of P. phocoena were not…”

L339-341
"...was significantly correlated with XXX, but was significantly correlated with YYY..." Something is wrong here. Either: A is significantly correlated with B and C; or A is significantly correlated with B but not C.


L342 “0.038, although after…”
“0.038; after…”

L353 “cochlear variation”
“those”

L357
What is moderate variation? Moderate relative to what?

L366 “biggest”
greatest, or largest

L379 “within specimens of…”
“among the specimens of…”

L384 (but see below)
Why don’t they refer to Fig. 1 here?

Annotated reviews are not available for download in order to protect the identity of reviewers who chose to remain anonymous.

·

Basic reporting

Overall, I am very happy with the paper as it stands, and the authors made all of my suggested revisions. I don't really have much in the way of critical comments, other than some minor grammatical corrections, which I will list below.

Experimental design

again, all of my major criticisms were answered, and the inclusion of information on there samples as well as the criteria that was used to define different ecological classifications has greatly improved the manuscript.

Validity of the findings

The inclusion of the suggested information above does not change the overall findings, which remain strongly supported and which suggest that intraspecific variation in cochlea morphology is relatively minor compared to that between different species.

Additional comments

While the paper is well written, I did find a couple of minor typos and issues.

First, I am not sure what the difference is between a specimen from the UK versus the North Atlantic. Doesn't the UK border the North Atlantic? If the authors could clarify what this difference means it would greatly improve interpretation. I assume "North Atlantic" might just mean "we are not sure where it comes from other than general region", but again, that is an assumption on my part.

Under "Data Collection", "large availability of specimens" should be "large number of specimens"

The authors should overall double check formatting. While it might be a relic of the edited word document, I will note that between 159 and 160 there is a random change in line spacing, There is a change in font size at line 404, and it looks like some paragraphs may have extra spacing or not enough spacing between them.

Line 450: "It is also possible our data contain some cryptic species." This reads as a throwaway statement with no elaboration. While I would definitely buy this line of argument for say, members of Tursiops and Delphinus, I have never heard of any suggestions that North Atlantic populations of Phocoena phocoena may include cryptic species. There is some evidence that populations in different ocean basins, such as between Pacific and Atlantic, could be distinct, but these shouldn't influence your results. I would delete this sentence

Figures: I will note that the caption for the PC Plot for diving in the supplemental material still has diving depths randomly distributed, rather than going very deep, deep, etc (or vice versa). Make more sense to order them by depth, as the brain will probably initially want to read it that way.

These are all minor elements and should be quick to fix. I otherwise fully endorse this paper, and I look forward to reading the finished version

---

## Round 0.3 · accepted · Accept

Thank you for working through the reviewer comments.